# The Changing Paradigm for the Treatment of HER2-Positive Breast Cancer

**DOI:** 10.3390/cancers12082081

**Published:** 2020-07-28

**Authors:** Aena Patel, Nisha Unni, Yan Peng

**Affiliations:** 1Department of Internal Medicine, Division of Hematology/Oncology, University of Texas Southwestern Medical Center, Dallas, TX 75390, USA; patel.aena@gmail.com; 2Department of Pathology, University of Texas Southwestern Medical Center, Dallas, TX 75390, USA

**Keywords:** HER2-positive breast cancer, metastatic disease, neoadjuvant and adjuvant therapy, targeted therapy, immunotherapy

## Abstract

For decades, HER2-positive breast cancer was associated with poor outcomes and higher mortality rates than other breast cancer subtypes. However, the advent of Trastuzumab (Herceptin) has significantly changed the treatment paradigm of patients afflicted with HER2-positive breast cancer. The discovery of newer HER2-targeted therapies, such as Pertuzumab (Perjeta), has further added to the armamentarium of treating HER2-positive breast cancers. This review highlights recent advancements in the treatment of HER2-positive diseases, including the newer HER2-targeted therapies and immunotherapies in clinical trials, which have paved (and will further update) the way for clinical practice, and become part of the standard of care in the neoadjuvant, adjuvant or metastatic setting.

## 1. Introduction

Breast cancer is categorized into four different molecular subtypes: Hormone receptor (HR)-positive (+)/Human epidermal growth factor receptor 2 (HER2)-negative (−) (Luminal A); HR+/HER2+ (Luminal B); HR−/HER2+ (HER2-enriched); and HR−/HER2− (triple negative). The survival rate among patients differ based on the molecular subtype and stage. The survival rate at four years among women with HR+/HER2− is estimated to be 92.5%, followed by HR+/HER2+ at 90.3%, HR−/HER2+ at 82.7%, and HR−/HER2− at 77.0% [1]. The HER2 (human epidermal growth factor receptor 2) oncogene is positive in about 20% of primary invasive breast cancers [2]. It is well known that HER2 overexpression is associated with higher rates of disease recurrence and mortality. In addition, HER2+ breast cancers have a higher predilection to metastasize to the brain. However, during the last two decades, the treatments and outcomes for patients with HER2+ disease have shifted dramatically. Trastuzumab (Herceptin) was first approved in 1998 as the first anti-HER2 directed therapy in metastatic HER2+ invasive breast cancer. Prior to trastuzumab, historically, patients with metastatic HER2+ disease were treated with traditional chemotherapy regimens. A pivotal Phase III trial [3] of 469 women showed that adding trastuzumab to standard chemotherapy (paclitaxel or anthracycline/cyclophosphamide) resulted in improved response rates (50% versus 32%), extended time to progression (7.4 months versus 4.6 months), and improvement in median overall survival (25 versus 20 months) [3]. Relative risk of death was also reduced by 20% at a median follow up of 30 months. Since trastuzumab, multiple agents have been developed to treat patients with HER2+ disease.

The HER superfamily consists of four tyrosine kinase receptors: HER1 (epidermal growth factor receptor), HER2 (neu, c-erbB2), HER3 and HER4. When activated, these receptors cause epithelial cell growth and differentiation. The HER2 oncogene encodes for a glycoprotein receptor with intracellular tyrosine kinase activity, and has no known ligand [4]. On the other hand, the other three HER receptors have known ligands, and form homodimers or heterodimers upon ligand binding, with the HER2 receptor being the preferred dimerization partner. The HER2 receptor can heterodimerize with the other receptors, which results in autophosphorylation of the tyrosine residues. This autophosphosphorylation activates the MAPK (mitogen-activated protein kinase) pathway and the PI3K (phosphatidylinositol 3-kinase) pathways. The HER2, or epidermal growth factor receptor 2, is the target for many HER2-directed therapies.

Herceptin binds to subdomain IV of HER2 in order to disrupt HER2 signaling. HER2 testing is done via immunohistochemistry (IHC), which tests for the overexpression of the HER2 gene product, and fluorescence in situ hybridization (FISH) to test for HER2 gene amplification. The tumor is identified as HER2+ if IHC is 3+ (intense staining within >10% of the tumor cells), or if the ratio of HER2 and the chromosome 17 enumeration probe (HER2/CEP17) is ≥2 and the HER2 copy number signals/cell equals is ≥ 4. It is important to note that those with non-HER2-overexpressing breast cancers do not derive benefits from adjuvant trastuzumab. This was studied in a randomized trial of 3270 patients [5] with invasive breast cancer, with IHC scores of 1+ or 2+ and with FISH <2 (or if the ratio was not performed, HER2 gene copy number <4.0). The study found that adding trastuzumab to chemotherapy (either docetaxel plus cyclophosphamide, or doxorubicin and cyclophosphamide, followed by weekly paclitaxel for 12 weeks) did not improve disease-free survival, distant recurrence-free interval, or overall survival.

This is in line with the updated 2018 ASCO/CAP (American Society of Clinical Oncology/College of American Pathologists) guidelines. The 2018 ASCO/CAP guidelines [6] identify a testing algorithm to address the less commonly found clinical scenarios, in order to address the infrequent HER2 results that are of unclear significance. Another major revision in the 2018 guidelines includes the revision of the definition of IHC 2+. IHC2+ is now defined as invasive breast cancer with weak to moderate complete membrane staining observed in >10% of tumor cells.

Here, we will address the key trials that have led to a major change in how we treat HER2+ invasive breast cancer in the neoadjuvant, adjuvant and metastatic settings. In addition, we will also discuss newer therapies, such as bispecific antibodies, and trials that are ongoing.

## 2. Neoadjuvant Treatment

Neoadjuvant HER2-based therapy is typically used (as in other subtypes) in locally advanced breast cancer (Stage IIb with T3 disease, or Stage III), or in patients with an earlier stage HER2+ disease who desire breast conserving therapy, have limited axillary nodal involvement (N1) (which could potentially be converted to node-negative disease and therefore result in sentinel lymph node biopsy), or have had surgery postponed (due to a variety of reasons). Given that pathologic complete response is associated with improved event free survival (EFS) (HR (hazard ratio) of EFS 0.37, 95% CI (confidence interval) 0.32–0.43) and overall survival (HR OS 0.34, 95% CI 0.26–0.42) [7], neoadjuvant therapy can help us gauge which patients are at higher risk of relapse, and more aggressive therapy can be offered. HER2-directed therapy is typically added to a chemotherapy backbone.

Several studies have demonstrated the efficacy of trastuzumab in improving event free survival, pathologic complete response (pCR), and overall survival. Based on a 2012 meta-analysis, herceptin increased pathologic complete response rates from 23% to 40% when added to neoadjuvant chemotherapy [8]. In addition, in the Phase III NOAH (Neoadjuvant Herceptin) trial [9], the addition of herceptin increased pCR rates from 19% to 38%, and improved event free survival (EFS) from 43% to 58% (HR 0.64, 95% CI 0.544–0.930). Given the improvement of event free survival at a median follow up 5 years (from trastuzumab-containing neoadjuvant therapy followed by adjuvant trastuzumab, in patients with locally advanced or inflammatory breast cancer), this trial highlights the association between pathologic complete response and long-term outcomes of patients with HER2+ disease.

Despite the improvement in pathologic complete response and event free survival, 15% of patients will relapse after therapy with trastuzumab due to a resistance to herceptin. The proposed mechanisms have included structural defects within the HER2 receptor [10], the constitutive activation of downstream elements [11], the activation of the downstream pathways by other members of the HER family [10,11], or intracellular alterations that affect the PI3K pathway [10]. Therefore, additional therapeutic targets with different mechanisms of action have been extensively studied in combination with trastuzumab, in order to evaluate whether these combination therapies prolong time to resistance and treatment failure [4].

Pertuzumab (Perjeta) is another biological therapy that has been studied in patients with HER2+ breast cancer. Pertuzumab is a monoclonal antibody that binds to subdomain II of the HER2 receptor and thereby blocks the heterodimerization with HER3, subsequently inhibiting downstream signaling. Given that pertuzumab enhances locoregional responses, it was approved in 2013 for patients with locally advanced, inflammatory, or early stage HER2+ invasive breast cancer (with size >2 cm or node-positive disease). The addition of pertuzumab to herceptin in the neoadjuvant setting was assessed in the NEOSPHERE Trial [12]. In the NEOSPHERE trial, the percentage of patients who achieved pCR was significantly higher in the pertuzumab + herceptin + docetaxel arm (46%) versus the herceptin + docetaxel arm (29%). The addition of pertuzumab led to increased diarrhea, however rates of cardiotoxicity were not higher with the combination of herceptin and pertuzumab, as assessed in the TRYPHAENA Trial [13]. Criticisms of the NEOSPHERE trial include the small sample size, the lack of patients from the United States, the lack of a blinded pathology review and the chosen chemotherapy backbone (which is not typically used in the United States).

If comorbidities preclude the addition of chemotherapy to targeted HER2 therapy, the other options include herceptin + pertuzumab, based on one arm of the NEOSPHERE trial where pcR rates were 16.8% (95% CI: 10.3–25.3, *p* = 0.0198) with pertuzumab + herceptin, versus herceptin + docetaxel. Another option includes the addition of lapatinib + herceptin (given for 18 weeks) based on the PAMELA trial [14]. Lapatinib is a small-molecule inhibitor of the tyrosine kinase inhibitors of both HER1 and HER2. In this trial, pathologically complete response rates were seen in 30% of the patients with previously untreated HER2+, or Stage I–IIIA breast cancer. Although this is an option for patients not wanting chemotherapy, this approach has not been adapted into practice.

Several cytotoxic regimens have proved efficacious in the neoadjuvant/adjuvant setting, including TCH +/− P (docetaxel, carboplatin, herceptin, pertuzumab), PCH +/− P (paclitaxel, carboplatin, herceptin, pertuzumab) or an anthracycline-based regimen, such as AC-TH +/− P (doxorubicin, cyclophosphamide followed by a taxane such as paclitaxel or docetaxel with herceptin +/− pertuzumab) or FEC/EC-TH +/− P (fluoruracil, epirubicin, cyclophosphamide, herceptin, pertuzumab). It is important to note that the safety of pertuzumab has not been established when combined with a doxorubicin-containing regimen. In addition, the safety of pertuzumab for more than six cycles in early-stage breast cancer has also not been established.

For those with low-risk disease, or patients with comorbidities, alternatives include weekly paclitaxel with herceptin (+/− pertuzumab) or a combination of docetaxel with cyclophosphamide, in addition to herceptin, every 3 weeks for four cycles [15], based on the results in the adjuvant setting.

## 3. Adjuvant Treatment

Adjuvant chemotherapy is given to patients with HER2+ disease that is a node-positive, or a node-negative disease with tumors >1 cm in size. After completion of chemotherapy and herceptin (given concurrently), the standard of care is to continue herceptin for a total of 52 weeks. Studies have found an improvement in overall survival, with a hazard ratio of 0.67 (95% CI 0.57–0.80) [16], when herceptin is administered for 12 months in the adjuvant setting. Extension to 2 years did not improve the 10-year disease-free survival, as studied in the HERA trial [17]. There was no reported difference in 10-year disease-free survival when herceptin was given for 1 year versus 2 years (HR 1.02, 95% CI 0.89–1.17). In addition, the study found that the incidence of cardiotoxicity was higher in the group who received herceptin for 2 years (7.3% versus 4.4%).

In contrast, a duration of less than 1 year of anti-HER2-directed therapy was proven to be more detrimental in the PHARE trial [18], which showed that treating patients with herceptin for 6 months resulted in more deaths, shorter 2-year disease-free survival rates, and more distant recurrences.

However, there is some data to support the shorter duration of HER2-directed therapy if patients cannot tolerate 12 months. In the recently published PERSEPHONE trial [19], patients with early stage HER2+ breast cancer were randomized to receive either 12 months or 6 months of adjuvant herceptin. Patients who received herceptin for 6 months had 4-year DFS rates similar to those who received adjuvant herceptin for 12 months (89.4% versus 89.8%, HR 1.07, 95% CI 0.93–1.24). It is important to note that the absolute difference in the 4-year DFS was only 0.4%. The difference between the discordant results between the PHARE and PERSEPHONE trials, despite a near equivalence of hazard ratios in both studies, has been attributed to the chosen non-inferiority margin of each trial [20]. The PHARE trial had a non-inferiority margin of 1.15. The upper bound of the two-sided 95% confidence interval was less than 1.15 in the PHARE trial. However, in the PERSEPHONE trial, the non-inferiority margin was defined as an absolute decrease in the 4-year DFS rate of 3%, which resulted in a non-inferiority margin of 1.316. In addition, the two-sided confidence interval was 90% in the PERSEPHONE trial, which thereby increasing the chance of concluding non-inferiority. Pondé et al. [20] note that if the non-inferiority margin of 1.15 was used in the PERSEPHONE trial, then the non-inferiority goal would not have been reached. In terms of safety, the PERSEPHONE trial demonstrated that patients who received 6 months of adjuvant herceptin experienced less cardiotoxicity (3% versus 8%, *p* < 0.0001) and fewer severe adverse events (19% in the 6-month group versus 24% in the 12-month group, *p* < 0.0002). However, it is important to note that 90% of the patients in the PERSEPHONE trial received anthracycline-based treatments. In addition, given that more non-anthracycline-based regimens are used in the current climate, the benefit of shortening the duration of trastuzumab to 6 months, in terms of cardiac safety, is not clear.

The KATHERINE trial evaluated the safety and efficacy of kadcyla (ado-trastuzumab emtansine or T-DM1) for 14 cycles vs. herceptin in the adjuvant setting, in patients with residual disease after neoadjuvant HER2-directed therapy [21]. Kadcyla is an antibody–drug conjugate of herceptin linked to an antimicrotubule agent (DM1). Patients with HER2+ early breast cancer with residual disease who received T-DM1 had an improved 3-year DFS, compared to receiving trastuzumab (88% versus 77%; HR 0.50, 95% CI 0.39–0.64). Although the number of serious adverse events was higher in the T-DM1 group (13% versus 8%), switching to T-DM1 in the adjuvant setting is associated with a lower risk of distant recurrence (HR 0.60, 95% CI 0.45–0.79).

The seven-year update of the single-arm, Phase II APT (adjuvant paclitaxel and trastuzumab) trial continued to demonstrate excellent outcomes, with disease-free survival of 93% and a recurrence-free interval (RFI) of 97.5% in small, node-negative HER2+ breast cancers. Based on these encouraging results, herceptin and paclitaxel for 12 weeks, followed by herceptin alone to complete 1 year of treatment, has emerged as a very tolerable and effective treatment option in this subset of patients [22].

Dual anti-HER2-directed therapy is recommended and approved for high-risk disease (node-positive or node-negative, with tumor size >2cm) in the adjuvant setting. In the Phase III APHINITY trial [23], adding pertuzumab to herceptin and chemotherapy led to an improvement in 3-year disease-free survival (94.1 versus 93.2%; HR 0.81, 95% CI 0.66–1.00), with subgroup analysis showing improvement in the patients with node-positive disease (92 vs. 90.2%; HR 0.77, 95% CI 0.62–0.96) but no difference in those with node-negative disease.

Various options exist for the chemotherapy backbone in the adjuvant setting. If an anthracycline-based regimen is used, the recommendation is to administer HER2-directed therapy sequentially, given the increased risk of cardiotoxicity. For example, we administer dose-dense doxorubicin + cyclophosphamide, followed by a taxane + herceptin (+/− pertuzumab). If a nonanthracycline regimen is given, preference is given to chemotherapy and anti-HER2-directed therapy concurrently.

## 4. Extended Adjuvant Treatment

Another agent, neratinib (dual kinase inhibitor that irreversibly inhibits the pan-Her receptors), was shown to improve recurrence rates when given after completion of 1 year of herceptin [24]. In the Phase III ExTENET trial, women with early-stage HER2+ disease were randomly assigned to receive neratinib or a placebo after treatment with herceptin. There was an improvement in 5-year invasive disease-free survival (90.2% vs. 87.7%; HR 0.73, 95% CI 0.57–0.92), with a subgroup analysis showing a more pronounced benefit in patients with hormone receptor-positive disease. Diarrheal prophylaxis is recommended, given the high frequency of grade 3–4 diarrhea. The role of neratinib in the post-KATHERINE trial era is unknown.

## 5. Metastatic Disease

In patients with HER2+ metastatic breast cancer, four HER2-directed therapies are approved (herceptin, pertuzumab, kadcyla or lapatinib). For patients who want to avoid chemotherapy, single agent herceptin can be used, however with the caveat that with progression, chemotherapy should be considered. For patients with hormone receptor-positive and HER2+ cancers, a combination of endocrine therapy and HER2-directed therapy can be used. However, if the disease is rapidly progressive or if there is visceral involvement, HER2-directed therapy plus chemotherapy is typically recommended.

Patients are typically treated with herceptin and a taxane (docetaxel or paclitaxel) + pertuzumab. Addition of pertuzumab was shown to improve overall response rates (80% versus 69%), progression-free survival (median PFS 19 versus 12 months) and overall survival (medial OS 56.5 versus 40.8 months) in the CLEOPATRA trial [25]. However, the addition of pertuzumab to herceptin and docetaxel led to an increased incidence of diarrhea, neutropenia, rash and serious febrile neutropenia, without increasing the risk of cardiotoxicity.

Cytotoxic chemotherapy is usually discontinued after 6–12 months if patients achieve a response. If patients progress six months after receiving herceptin, triple therapy with herceptin, pertuzumab and a taxane can be re-introduced. However, if patients progress within six months of receiving herceptin, kadcyla (ado-trastuzumab emtansine) is recommended. In the Phase III EMILIA trial [26], patients who were previously treated with herceptin and a taxane were randomized to receive kadcyla or lapatinib + capecitabine. There was an improvement in progression-free survival (10 months vs. 6 months, HR 0.65, 95% CI 0.55–0.77), overall survival (median OS 31 months vs. 25 months; HR 0.68, 95% CI 0.55–0.85) and overall response rate (44% vs. 31%). Serious toxicities included thrombocytopenia (13% versus 0.2%) and a higher incidence of bleeding (30% versus 16% in the lapatinib + capecitabine arm). In the TH3RESA trial [27], for patients with unresectable, locally advanced, metastatic or recurrent HER2+ breast cancer, who had progressed while on two HER2-directed therapies (herceptin and lapatinib), kadcyla resulted in an improvement in progression-free survival (6.2 vs. 3.3 months; HR 0.53, 95% CI 0.42–0.66) and improvement in overall survival (22.7 vs. 15.8 months; HR 0.68, 95% 0.52–0.85). Given the data, kadcyla is usually considered in the second line setting. The Phase III MARIANNE trial [28] studied kadcyla in the first line setting. The study found that there was no significant difference between herceptin with a taxane versus kadcyla with a placebo versus kadcyla with pertuzumab. Upon progression while on kadcyla and trastuzumab-containing regimens, other options, such as lapatinib + capecitabine (given the improvement in time to progression compared to capecitabine alone), or a combination of anti-hormonal therapy along with anti HER2 therapy, can be considered. In the metastatic setting, HER2-directed therapy is often continued even with disease progression.

## 6. Brain Metastasis in HER2+ Disease

HER2-targeted therapies could be considered instead of locally directed therapies, such as radiation, in patients with brain metastasis from HER2+ breast cancer. After progression while on trastuzumab (with or without pertuzumab) with a taxane, trastuzumab-emtansine is typically utilized, based on the retrospective exploratory analysis from the EMILIA trial [29]. Among patients with CNS (central nervous system) metastasis, there was significant improvement in overall survival in the T-DM1 arm, compared to the lapatinib + capecitabine arm (hazard ratio of 0.38, *p* = 0.008, 26.8 months versus 12.9 months). The efficacy of capecitabine + lapatinib was studied in the LANDSCAPE trial [30]. In this trial, 66% of the patients (29 patients out of 44 patients) were found to have a partial response, with a median time to progression of 5.5 months. The 6-month overall survival was 90%. The combination of lapatinib and capecitabine is typically utilized as a later-line therapy.

The next line of therapy for patients with HER2+ CNS metastasis, after progression while on kadcyla, is typically a combination of tucatinib, capecitabine and trastuzumab [31]. Specifically in relation to brain metastases, 25% of the patients with brain metastases had a one-year PFS when treated with tucatinib with capecitabine and trastuzumab, compared to 0% in the trastuzumab + capecitabine arm (HR 0.48, 95% CI 0.34–0.69). The median PFS in the tucatinib arm was 7.6 months versus 5.4 months in the capecitabine + trastuzumab arm.

Another option for patients with HER2+ metastases is neratinib + capecitabine, based on the Phase II trial [32] that studied 49 patients with HER2+ brain metastasis. Among the lapatinib naïve patients, the objective response rate was 49%, with a median PFS of 5.5 months and overall survival 13.3 months. Among patients who received lapatinib previously, the objective response rate was 33%, with a median PFS of 3.1 months and overall survival of 15.1 months.

Bevacizumab, a monoclonal antibody against vascular endothelial growth factor A, is a systemic therapy option that has been studied in single-arm, Phase II studies [33], with reported response rates of >50% when added to a platinum-based regimen with cisplatin and etoposide.

## 7. Mechanism of Action of HER2-Directed Therapies and Resistance Mechanisms

As discussed above, the HER2 receptor is a transmembrane tyrosine kinase receptor that belongs to the human epidermal growth factor receptors (EGFR). It is expressed at a low level on the surface of epithelial cells, and is needed for development in several tissue types, such as the breast, ovary, central nervous system, lung, liver and kidney [10]. It is overexpressed in 25–30% of breast cancer cells. As shown in Figure 1, HER2 forms homodimers (binding of same receptor) or heterodimers (binding of different receptors) with other members of the human epidermal growth factor receptors. The HER2 protein can exist in an inactivated state, and dimerize independent of the binding of a ligand. The binding of a ligand induces phosphorylation of the receptors, which in turn activates the MAPK (mitogen-activated protein kinase) pathway and the PI3K (phosphatidylinositol 3-kinase) pathways.

The development of trastuzumab revolutionized the treatment of HER2+ breast cancer by introducing a monoclonal antibody that specifically targeted breast cancer cells that overexpressed aberrant HER2 receptors. Trastuzumab binds to the domain IV region of the extracellular site of the HER2 protein, thereby preventing dimerization, and subsequently signal transduction and cell survival. Pertuzumab is a monoclonal antibody that binds to domain II of the extracellular component of HER2, thereby preventing dimerization with Her1 and HER3. Ado-trastuzumab-DM1 (T-DM1, Kadcyla) combines the antibody (trastuzumab) with DM1 (anti-microtubule agent derived from maytansine), which then delivers the drug in the intracellular compartment [34]. Trastuzumab, pertuzumab and T-DM1 utilize antibody-dependent cellular cytotoxicity. Lapatinib is a dual tyrosine kinase inhibitor that reversibly binds to the tyrosine kinase receptors (EGFR or Erb1, and HER2 or ErbB2), thereby blocking the phosphorylation and activation of ERK (extracellular signal-regulated kinase) and AKT (protein kinase B). Neratinib, on the other hand, is an irreversible tyrosine kinase inhibitor of HER1, HER2 and HER4, thereby preventing the downstream signaling of the MAP kinase pathway and the AKT signaling pathways.

Although these HER2-targeted therapies have been proven to be efficacious in the various studies previously mentioned, studies have found that less than 35% of patients with HER2+ breast cancer initially respond to trastuzumab [10]. On the other hand, some patients acquire resistance after being on HER2-targeted therapies for several months after initial response. Several mechanisms of resistance to the trastuzumab-based therapy have been proposed [10]. Dr. Vu and colleagues [10] have shown that defects within the HER2 receptor, such as a truncated extracellular domain, prohibits the binding of trastuzumab to the receptor. For example, HER2 can mutate in a fashion that results in a truncated p95HER2 isoform, which inhibits the binding of trastuzumab due to the lack of the extracellular domain [35]. Studies have found that those who acquired the p95HER2 mutation were less likely to respond to trastuzumab as this mutated isoform results in constitutive kinase activity [36]. One way to overcome this form of resistance is to add lapatinib to trastuzumab, given that lapatinib acts intracellularly [37].

Another broad category in the different mechanisms of resistance includes elevations of other tyrosine kinase inhibitors. For example, studies have found that the cross-signaling between Insulin-like Growth factor-1 receptor (IGF-IR) and HER2 induces phosphorylation of HER2, and thereby activates signaling transduction [38]. Overexpression of c-met (tyrosine protein kinase met or hepatocyte growth factor receptor) has been found to confer resistance to trastuzumab [39].

Intracellular alterations can also cause resistance to HER2-targeted therapy. One such alteration involves the hyperactivation of the PI3K pathway by mutations or loss of PTEN (phosphatase and tensin homolog deleted on chromosome 10), which is a tumor suppressor gene that normally inhibits the PI3K pathway [35]. Patients with PTEN-deficient tumors had lower overall response rates to trastuzumab than patients with wild type PTEN [40]. Clinical trials, incorporating different drugs and combination strategies to overcome these mechanisms of resistance, are in progress.

Not much is known about the mechanisms of resistance to pertuzumab. However, one theory that has been proposed is the suppression of microRNA-150, which are small, noncoding, single-stranded RNAs, that negatively regulate the PI3K-AKT pathways. Studies in ovarian cancer [41,42] have found that the suppression of miRNA-150 resulted in decreased sensitivity to pertuzumab.

Given that combining trastuzumab with lapatinib targets both the intracellular and extracellular HER2 domains [37], combining these two drugs is an attractive strategy. However, acquired mechanisms of resistance to lapatinib often develop after chronic exposure to lapatinib. One theory is that lapatinib promotes the transcription of estrogen-positive genes, and therefore switches cell survival dependence from HER2 to estrogen receptors [43]. Another proposed mechanism involves the activation of AXL, a membrane-bound tyrosine kinase [44]. AXL has been associated with activation of the AKT/MTOR pathway [37].

The proposed theories concerning resistance to T-DM1 (kadcyla) include low tumor HER2 expression, poor internalization of the HER2-T-DM1 complexes, defective intracellular trafficking of the HER2-T-DM1 complex, and defective lysosomal degradation of T-DM1. These result in inadequate drug concentrations, and therefore cell death is halted [45]. Another proposed mechanism of resistance to T-DM1 is the presence of neuregulin b1 (NRG), which suppresses the cytotoxic activity of T-DM1 by triggering the formation of HER2-HER3 heterodimers. This heteromization activates the PI3K pathway, and thus leads to cell cycle proliferation independently of kadcyla [46,47]. SYD985, another HER2-targeting antibody-drug conjugate [48], is currently being studied in patients who develop resistance to T-DM1.

## 8. Newer HER2-targeted Therapies

Although trastuzumab (herceptin), pertuzumab (perjeta), Lapatinib (tykerb) and ado-trastuzumab emtansine (kadcyla) remain the most used HER2-targeted therapies in practice, other new HER2 therapies are being studied in clinical trials.

In the Phase III SOPHIA trial [49], margetuximab was compared to trastuzumab in patients with metastatic HER2+ breast cancer after progression while on the first, second or third line of therapy including kadcyla. In the study, the median age of the patients was 55 years. The backbone chemotherapy was of the investigator’s choosing, between four systemic cytotoxic therapies (capecitabine—27% in each arm; eribulin—25% in each arm; gemcitabine—12% in each arm; vinorelbine—33% in each arm). Of note, more than 90% of the patients received kadcyla.

Margetuximab is a novel Fc-engineered monoclonal antibody that targets the HER2 oncoprotein. The Fc portion enhances the immune system in order to provide an added benefit. Of the patients in the study, 85% carried the CD16A 158F allele, which is shown to have a diminished response to trastuzumab. The patients who were homozygous for the CD16A-F allele appeared to attain longer progression-free survival with margetuximab compared to trastuzumab. Patients treated with margetuximab had a higher median PFS and higher overall response rates. The adverse reactions were similar between the two groups.

In the ALTERNATIVE trial [50], the addition of lapatinib to the trastuzumab and endocrine therapy (aromatase inhibitor) was studied in postmenopausal women with HER2+, hormone receptor-positive metastatic breast cancer. The addition of lapatinib to trastuzumab led to higher median progression-free survival (primary end point) and overall survival (see Table 1) rates.

The median progression-free survival associated with the combination of lapatinib with trastuzumab was 11 months, versus 5.7 months in the trastuzumab arm (HR 0.62, 95% CI 0.45–0.88). The data for overall survival was immature in the trial. The addition of lapatinib (and removal of pertuzumab) could be considered in patients who progressed after receiving trastuzumab, pertuzumab and an AI, if the disease is not rapidly progressive or if there is not a visceral crisis.

As of May 2020, there are two novel agents approved for the treatment of patients with advanced HER2+ breast cancer, and who have previously been treated.

In December 2019, the FDA approved trastuzumab deruxtecan-nxki (DS-8201a/Enhertu) in patients who have been previously treated with two or more prior anti-HER2 therapies in the metastatic setting. This approval was based on the results of an open-label, single group, phase 2 study with 184 patients who had previously received a median of six lines of therapy [52]. The primary end point of the DESTINY- Breast01trial (Available online: https://clinicaltrials.gov/ct2/show/NCT03248492, identifier NCT03248492, last accessed 7/12/2020) was an overall response rate observed to be 60.9%, with patients achieving a disease control rate of 97.3%, median progression-free survival of 16.4 months and a median duration of response of 14.8 months. The median overall survival has not been reached. DS-8201 is an antibody–drug conjugate like kadcyla, but with a topoisomerase I inhibitor payload linked to a humanized monoclonal antibody, and to HER2, by a cleavable tetrapeptide linker. The most common adverse events noted were neutropenia, interstitial lung disease, anemia, nausea and alopecia. Death due to interstitial lung disease was reported in 2.2% of patients. Currently, ongoing trials are evaluating the effectiveness of DS-8201a against kadcyla in a randomized control trial (DESTINY-Breast03 trial; NCT03529110), as well as the effectiveness of capecitabine and herceptin or lapatinib (DESTINY-Breast02 trial; NCT03523585).

Based on the results of the HER2CLIMB trial [31], another novel HER2-targeted agent, tucatinib, was approved in April 2020 [31]. Tucatinib is an orally bioavailable, highly selective small-molecule HER2 tyrosine kinase inhibitor, which binds to the internal domain of the HER2 protein. Given the small size of the molecule, tucatinib is believed to cross the blood–brain barrier. It causes minimal EGFR inhibition, and hence is relatively well tolerated. Patients with previously treated, unresectable, locally advanced or metastatic HER2+ breast cancer were treated with either tucatinib + trastuzumab + capecitabine or with placebo + trastuzumab + capecitabine. The study allowed patients with untreated but asymptomatic brain metastasis to participate. Tucatinib combination therapy showed statistically significant improvements in progression-free survival at 1 year (primary end point), as well as in the overall survival at 2 years, in the overall population.

Given the high response rate, tucatinib combination is a promising therapy for previously treated HER2+ metastatic breast cancer patients, and could emerge as the go-to option, especially in patients with central nervous system involvement.

Neratinib is a pan-HER inhibitor, which has been recently approved in the metastatic setting, based on the NALA trial [53] and the TBCRC 022 [32]. In the Phase III NALA trial, neratinib plus capecitabine improved progression-free survival compared to lapatinib with capecitabine (12-month PFS was 29%, versus 15%). [53]. However, patients who received neratinib with capecitabine experienced more diarrhea. 25% of the patients in the neratinib plus capecitabine group experienced grade > 3 diarrhea compared to 13% of the patients receiving lapatinib with capecitabine.

Pyrotinib and poziotinib are both irreversible pan-HER inhibitors. Pyrotinib has already received conditional approval in China, based on a Phase II trial which showed improved PFS in combination with capecitabine, with tolerable side effects [54]. A Phase III trial is currently ongoing (NCT03080895).

## 9. Novel Antibody Drug Conjugates (ADCs)

Following the success and FDA approval of two Antibody Drug Conjugates (ADCs), there are multiple other agents currently in clinical trials. Like DS-8201, these newer ADCs have a cleavable linker that accounts for what is called the “bystander effect”. Bystander effect is responsible for the death of antigen-negative cells (both cancer cells and normal cells), hence it is important in both the efficacy and safety of the drug. Here we discuss a few ADCs with published clinical trial results.

These second-generation ADCs are thought to overcome the resistance of HER2+ cells to T-DM1. One of the ADCs currently in clinical trial, SYD985 (Trastuzumab Duocarmazine), was shown to be effective in T-DM1-resistant patient-derived tumor models [55]. Results of the dose-escalation/dose-expansion study with this agent demonstrated clinical activity in heavily pre-treated HER2+ patients, with a partial response of 33% and a median PFS of 7.6 months [55]. Currently, the Phase III randomized TULIP trial (NCT03262935) is evaluating the drug SYD985 against other standard of care options for previously treated HER2+ breast cancer patients.

Another ADC currently in Phase III trial is BAT8001 (Bio-Thera; NCT04185649), which uses a novel non-cleavable linker between trastuzumab and the maytansine payload. A phase I dose-escalation study (NCT04189211) revealed the drug to be safe, and it also showed efficacy in heavily pre-treated HER2+ patients. A randomized multi-center Phase III trial is ongoing in China [56].

RC48-ADC, or Distamab vedotin (RemeGen), is another antibody–drug conjugate with a cleavable cathepsin linker attached to the monomethyl auristatin E (MMAE) payload. In the Phase Ib/II trial, RC48 demonstrated good tolerability and efficacy. The disease control rate was seen to be 96.7%, with a 46.7% clinical benefit rate (CBR) [57]. A Phase II study is ongoing (NCT03500380).

DHES0815A (Genentech) is an engineered ADC, in which trastuzumab is conjugated with a stable linker to a highly toxic payload, pyrrolobenzodiazepine, and is the subject of an ongoing Phase I study (NCT03451162).

## 10. Bispecific Antibodies

These are molecules that recognize two different epitopes of the HER2 protein, and there are several of these currently in trials. In addition to these antibodies blocking tumor signaling pathways, they also engage immune cells and deliver payloads to destroy tumor cells [58].

ZW49 (Zymeworks) is a bispecific, biparatopic antibody, with an auristatin payload of an anti-HER2 biparatopic antibody, ZW25, which binds the same domains as trastuzumab (ECD2) and pertuzumab (ECD4). In pre-clinical breast cancer cell lines and PDX models, ZW49 has demonstrated anti-tumor activity [59]. A Phase I, dose-finding, multicenter, open-label trial is ongoing to assess the safety and tolerability of the drug (NCT03821233).

ZW25 increases tumor cell binding, improves receptor internalization and downregulates HER2 expression. A Phase I study showed a partial response rate of 33%, with a disease control rate of 50% [60].

BTRC4017A (Genentech) is currently being explored in a Phase Ia/b clinical trial (NCT03448042). It has a T-cell-dependent bispecific monoclonal antibody with two antigen recognition sites, one for HER2 and another one for the CD3 complex, which leads to the cross linking of HER2-expressing tumor cells and cytotoxic T lymphocytes.

TrasGex or Timigutuzumab is a glyco-optimized antibody, which has shown efficacy in a dose-escalation Phase I study by enhancing the antibody-dependent cell-mediated cytotoxicity [61]. However, it is unknown if the molecule is still being developed or not.

The experimental drug, NJH395 (Novartis), an immunoconjugate immune stimulator antibody conjugate (ISAC) consisting of a monoclonal anti-ErbB2 antibody conjugated to a TLR7 (Toll-like receptor 7) agonist, is another drug currently being studied in clinical trials (NCT0369771). No pre-clinical data is available yet.

## 11. Immunotherapy in HER2+ Disease

In addition to newer anti-HER2-targeted therapies, immunotherapy is being extensively studied given the ability of cancers to evade the immune system [62]. Different modalities are being utilized in immunotherapy to boost the patient’s immune system so as to attack the cancerous cells. Immune checkpoint inhibitors, such as cytotoxic T-lymphocyte antigen-4 (CTLA-4) antibodies and programmed cell death-1 (PD-1)/programmed cell death-1 ligand (PD-L1) antibodies, are being studied in clinical trials. The expression of PDL-1 has been shown to be associated with unfavorable characteristics, such as HER2+ status in addition to large tumor sizes and high tumor grades [63]. Although PDL-1 therapy is approved for triple negative breast cancers [64], there is no approved PD-1/PDL-1 agent for the treatment of HER2+ breast cancers. In the Javelin Solid Tumor study (Phase Ib) [65], avelumab had an overall response rate of 3% (of the 168 patients in the study; 26 (15.5%) patients were HER2+), but a higher activity was seen if they exhibited PDL-1 expression. Most recently, in the single-arm, multicenter PANACEA trial [66], pembrolizumab was studied in patients with advanced HER2+ breast cancer, who progressed while on the trastuzumab or T-DM1 therapy in this Phase Ib/II trial. While six patients were enrolled in the phase Ib trial (all PDL-1-positive), 52 patients were enrolled in the phase II trial. Patients were tested for PDL-1 expression. Of the 52 patients in the phase II trial, 40 had PDL-1-positive tumors, and the remaining patients had PDL-1-negative tumors. Although no objective responses were seen in the PDL-1-negative patients, the disease control rate was 24% in PD-L1-positive patients, and the overall response rate was 39%. The most common adverse reaction that was noted was fatigue (21%).

## 12. Future Directions

Although patients do relatively well on the therapies previously mentioned, a subset of HER2+ breast cancer patients experienced relapse after neoadjuvant and adjuvant therapy, as well as resistance to existing therapies. A dual-targeted HER2 approach is typically used if there are no contraindications. However, what remains to be determined is whether adding a newer HER2-targeted therapy would increase progression-free survival, or whether immunotherapy would help in overcoming the resistance to HER2-targeted therapies. In addition, many trials did not utilize the updated 2018 ASCO (American Society of Clinical Oncology)/CAP (College of American Pathologists) guidelines on HER2 interpretation in breast cancer [6], which generated an increase in negative cases due to the more rigorous algorithm for identifying HER2+ patients. This algorithm helped to reclassify equivocal cases as either HER2+ or HER2−. Treatment of HER2 equivocal cases was not standardized, given the lack of data surrounding the clinical benefits of HER2-targeted therapies in this subset of patients. Since the 2018 guidelines reclassify these patients as either HER2+ or HER2−, the overtreatment of patients can be avoided, and costs can be saved.

There is still an unmet need for treatment of metastatic HER2+ breast cancer. Whether adding or utilizing immunotherapy upfront, like in other subtypes of breast cancer, can improve patient outcomes remains unknown. Different strategies are being employed to improve the efficacy of anti-HER2 therapy by combining existing approved therapies and exploring next-generation sequencing to target potential biomarkers, so as to overcome resistance and reduce side effects. There is much more to be studied in HER2+ breast cancer.

## 13. Conclusions

The discovery and implementation of new HER2-directed therapies in clinical practice has significantly changed how patients with HER2+ diseases are being treated. Although major strides have been made in treating patients with HER2+ diseases, studies are ongoing concerning the continued improvement of the outcomes for this subset of patients. Whether it is a combination of multiple HER2-directed therapies in the various settings, or the invention of immunotherapy, the treatment of HER2+ disease has resulted in better outcomes, including in progression-free survival and overall survival, compared to the outcomes of previous decades, and these will continue to evolve.

## Figures and Tables

**Figure 1 cancers-12-02081-f001:**
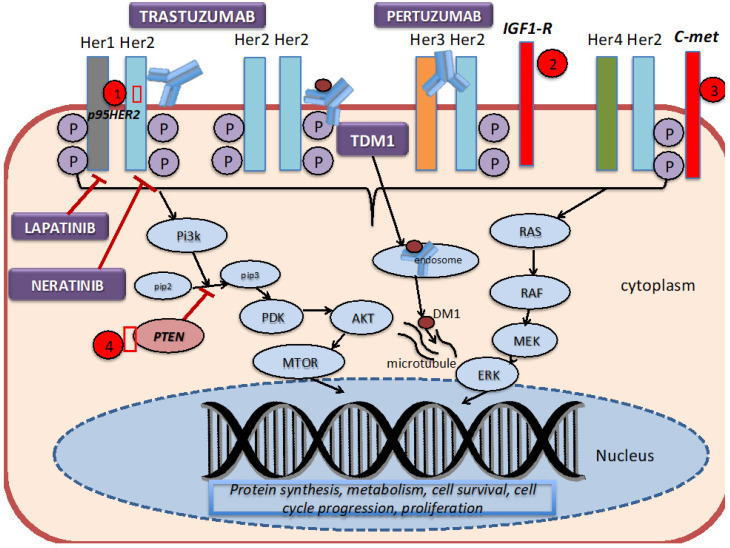
HER2 signaling pathway, mechanism of action of targeted therapies, and resistance mechanisms. 1. The truncated P95HER2 isoform results in the loss of the extracellular binding site for trastuzumab. 2.–3. Overexpression of other tyrosine kinase receptors, such as IGF1-R and C-met, can continue to trigger downstream signaling despite blockade by trastuzumab. 4. Mutations or loss of PTEN constitutively activates the PI3K signaling pathway.

**Table 1 cancers-12-02081-t001:** Trials of Newer HER2 Targeted Therapies.

Trial	Patients and Key Inclusion Criteria	Study Design	Results	Adverse Events
Phase III SOPHIA trial [49]	*n* = 536 Pre-treated (lines 1–3) HER2+ metastatic BC	MARG 15 mg/kg q3weeks vs. TRAS 8 mg/kg loading dose followed by 6 mg/kg q3w + investigator’s choice (capecitabine, eribulin, gemcitabine or vinorelbine)	PFS: (HR 0.76, *p* = 0.033)-higher PFS if homozygous for CD16A-F allele Median PFS: 5.8 mo (MARG) vs. 4.9 months (TRAS) ORR: 22.1% (MARG) vs. 16.0% (TRAS); *p* = 0.060	Infusion reaction: 12.9% (MARG) vs. 3.8% (TRAS) Adverse events of any grade were similar between MARG and TRAS
Phase III ALTERNATIVE Trial, adding lapatinib to herceptin and aromatase inhibitor [50]	*n* = 355 Postmenopausal women with HER2+, HR+ MBC (had received prior ET and prior neoadjuvant or first line TRAS + chemo)	1:1 randomization LAP 1000 mg/d + TRAS (*n* = 120) + AI vs. LAP 1500 mg/d + AI (*n* = 118) vs. TRAS + AI (*n* = 120) AI: letrozole 2.5 mg/d, anastrozole 1 mg/d or exemestane 25 mg/d	Median PFS: 11.0 mo (LAP + TRAS + AI) vs. 5.7 mo (TRAS + AI) (HR = 0.62, *p* = 0.0064) vs. 8.3 mo (LAP + AI) (HR = 0.71, *p* = 0.361) ORR: 31.7% (LAP + TRAS + AI) vs. 13.7% (TRAS + AI) vs. 18.6% (LAP + AI)	
Phase IB HER2 CLIMB (TUC) trial [51]	*n* = 60 HER2+ metastatic BC, including patients with untreated or progressive brain metastasis	Not randomized TUC (300 mg bid) + CAP vs. TUC + TRAS vs. TUC +CAP + TRAS	RR: 42% (5/12) in patients with brain mets (TUC + CAP +TRAS) ORR: 61% (14/23) in the triple regimen Median duration of response: 11.0 (range, 2.9–18.6) in triplet regimen	Grade 1–2 in triplet regimen: diarrhea (33%), nausea (26%) and fatigue (15%) Dose-limiting toxicity: grade 4 cerebral edema in a patient with untreated brain metastasis who was not on steroids

BC, breast cancer; MARG, margetuximab; TRAS, trastuzumab; HR, hormone receptor; MBC, metastatic breast cancer; ET, endocrine therapy; LAP, lapatinib; AI, aromatase inhibitor; ORR, overall response rate; OS, overall survival; PFS, progressive free survival; TUC, tucatinib; CAP, capecitabine; RR, response rates.

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
