# Peer review of "The Changing Paradigm for the Treatment of HER2-Positive Breast Cancer"

_cancers, 2020, doi:10.3390/cancers12082081_

Round 1

Reviewer 1 Report

The authors have improved the quality of the manuscript but there are still some comments that need to be addressed.

Line 28: “…HR-/Her2+ is estimated at 92.5%,…”. Do the authors mean “HR+/Her2-”?

Line 51: “PI3K (phosphatidylinositol 3-kinase) pathways..” Please remove one of the full stops.

Line 54: “Immunohistochemistry” should be lower case.

Line 65: “The recent ASCO 2018 Guidelines…”. This sentence is unclear. Is it in relation to the need for a better classification of Her2+ breast cancer? Or do the authors refer to the new 2018 treatment recommendations for Her2+ patients? Please modify.

Line 91: “…15% of patients of relapse…” should be “15% of patients will relapse…”

Line 133: “the standard of care is continue…” should be “the standard of care is to continue…”

Line 150: “has been been” should be “has been”

Lines 165 and 217: “emastine” should be “emtansine”.

Line 193: “…was shown to improvement in recurrence rates” should be “…was shown to improve recurrence rates”.

Line 240: “restrspective” should be “retrospective”.

Line 259: What is Bevacizumab? Please give more details.

Line 291: Do the authors mean “Dr Vu and colleagues demonstrated…”?

Line 331: “T-DM—1” should be “T-DM1”.

Section 11 “Newer Tyrosine Kinase Inhibitors” should be combined with section 8 “Newer Her2 Targeted Therapies”.

Page 11: When the authors refer to data from clinical trials that have not been published, they should indicate the clinical trial ID number.

Line 434: is NJH395 being tested in clinical trials? Are there any pre-clinical data available? Please comment.

Lines 472-473:”In addition, many trials did not utilize…” How does this impact on previous trials and how will it be addressed in future studies?

Lines 475-476: “Whether adding or utilizing immunotherapy upfront like in other subtypes of breast cancers remains unknown.” Do the authors mean: “Whether adding or utilizing immunotherapy upfront like in other subtypes of breast cancers can improve patients’ outcome remains unknown”?

The authors should use the same abbreviations consistently throughout the manuscript. E.g., Her2+ vs Her2 positive, T-DM1 vs TDM-1 … Please carefully check the entire manuscript. Similarly, for the names of the clinical trials that are sometimes in capital letters and sometimes not.

Author Response

Reviewer #1

The authors have improved the quality of the manuscript but there are still some comments that need to be addressed.

Line 28: “…HR-/Her2+ is estimated at 92.5%,…”. Do the authors mean “HR+/Her2-”?

Yes, it’s been changed to HR+/Her2-.

Line 51: “PI3K (phosphatidylinositol 3-kinase) pathways..” Please remove one of the full stops.

Removed

Line 54: “Immunohistochemistry” should be lower case.

Changed to lower case immunohistochemistry

Line 65: “The recent ASCO 2018 Guidelines…”. This sentence is unclear. Is it in relation to the need for a better classification of Her2+ breast cancer? Or do the authors refer to the new 2018 treatment recommendations for Her2+ patients? Please modify.

The updated 2018 ASCO (American Society of Clinical Oncology)/CAP (College of American Pathologists) guidelines [6] identifies a testing algorithm to address the less commonly found clinical scenarios in order to address the infrequent HER2 results that are of unclear significance. Another major revision in the 2018 guidelines includes the revision of the definition of IHC 2+. IHC2+ is now defined as invasive breast cancer with weak to moderate complete membrane staining observed in >10% of tumor cells.

Line 91: “…15% of patients of relapse…” should be “15% of patients will relapse…”

changed to “will”

Line 133: “the standard of care is continue…” should be “the standard of care is to continue…”

changed

Line 150: “has been been” should be “has been”

removed the extra been

Lines 165 and 217: “emastine” should be “emtansine”.

Both changed to emtansine

Line 193: “…was shown to improvement in recurrence rates” should be “…was shown to improve recurrence rates”.

changed

Line 240: “restrspective” should be “retrospective”.

Changed to retrospective

Line 259: What is Bevacizumab? Please give more details.

Bevacizumab, a monoclonal antibody against vascular endothelial growth factor A, is a  systemic therapy option

Line 291: Do the authors mean “Dr Vu and colleagues demonstrated…”?

Dr. Vu and colleagues [10] have shown that defects within the HER2 receptor such as a truncated extracellular domain prohibits the binding of trastuzumab to the receptor.

Line 331: “T-DM—1” should be “T-DM1”.

Changed to T-DM1

Section 11 “Newer Tyrosine Kinase Inhibitors” should be combined with section 8 “Newer Her2 Targeted Therapies”.

Combined the two sections

Page 11: When the authors refer to data from clinical trials that have not been published, they should indicate the clinical trial ID number.

Clinicaltrials.gov identifier IDs added to all the clinical trials that are not published yet

Line 434: is NJH395 being tested in clinical trials? Are there any pre-clinical data available? Please comment.

The experimental drug, NJH395 (Novartis), an Immunoconjugate Immune stimulator antibody conjugate (ISAC) consisting of a monoclonal anti-ErbB2 antibody conjugated to a TLR7 agonist, is another drug currently being studied in clinical trials (NCT0369771). No pre-clinical data is available yet.

Lines 472-473:”In addition, many trials did not utilize…” How does this impact previous trials and how will it be addressed in future studies?

In addition, many trials did not utilize the updated 2018 ASCO (American Society of Clinical Oncology)/CAP (College of American Pathologists) guidelines on HER2 interpretation in breast cancer [6], which generated an increase in negative cases due to a more rigorous algorithm to identify HER2+ patients. This algorithm helped to reclassify equivocal cases as either HER2+ or HER2-. Treatment of HER2 equivocal cases was not standardized given the lack of data surrounding the clinical benefit of HER2 targeted therapies in this subset of patients. Since the 2018 guidelines reclassifies these patients as either HER2+ or HER2-, overtreatment of patients can be avoided and costs can be saved.

Lines 475-476: “Whether adding or utilizing immunotherapy upfront like in other subtypes of breast cancers remains unknown.” Do the authors mean: “Whether adding or utilizing immunotherapy upfront like in other subtypes of breast cancers can improve patients’ outcome remains unknown”?

Changed to the latter sentence

The authors should use the same abbreviations consistently throughout the manuscript. E.g., Her2+ vs Her2 positive, T-DM1 vs TDM-1 …

Changed all variations of Her2 positive to HER2+ (except for the abstract and title). All incorrect spellings for T-DM1 changed to T-DM1.

Please carefully check the entire manuscript. Similarly, for the names of the clinical trials that are sometimes in capital letters and sometimes not.

Changed all trial names to all capital trials

Reviewer 2 Report

Authors have revised the manuscript to satisfy all comments. 

A minor comment before publication: There are some unusual symbols (question marks) that appear in Figure 1. This need to be fixed before publication. 

Good luck! 

Author Response

Reviewer #2

Authors have revised the manuscript to satisfy all comments. 

A minor comment before publication: There are some unusual symbols (question marks) that appear in Figure 1. This need to be fixed before publication. 

Not sure if it’s the formatting on their end but I cannot see these question mark symbols in the figure.

Good luck! 

Round 2

Reviewer 1 Report

The authors have adequately addressed my comments and suggestions.

Minor comments:

Lines 64-69: This paragraph seems disconnected from the previous one. My suggestion: ”This is in line with the updated 2018 ASCO…”

Line 450: Could the authors define “TLR7”?

Question marks still appear all over Figure 1.

Author Response

The authors have adequately addressed my comments and suggestions.

Minor comments:

Lines 64-69: This paragraph seems disconnected from the previous one. My suggestion: ”This is in line with the updated 2018 ASCO…”

Added the sentence to improve the transition.

Line 450: Could the authors define “TLR7”?

Toll-like Receptor 7

Question marks still appear all over Figure 1.

We don’t see question marks from our end for some reason. Not sure if it’s related to the formatting. I added more spaces in between the paragraphs and the figure in the hopes that these question marks disappear. I’m not sure what else to do because I can’t see them in my word document.

This manuscript is a resubmission of an earlier submission. The following is a list of the peer review reports and author responses from that submission.

Round 1

Reviewer 1 Report

The aim of this review is to provide an overview of clinical trials in patients with early stage, locally advanced and metastatic HER2 positive breast cancers. In all sections, the authors should avoid making simple lists of the different clinical trials for HER2 positive breast cancer patients and rather discuss how results from these trials provide new insight into the most effective treatment regimens that can lead to implementation of new therapeutic strategies in clinical practice.

My specific comments are below.

Introduction: The authors should consider revise the structure of the introduction by stating briefly, how the patients with HER2 positive breast cancer used to be treated before the introduction of HER2 targeted therapies and how they are treated now. The authors should also explain what HER2 is, the downstream signalling pathways it activates and the different types of inhibitors that have been developed so far. Finally, the authors should clarify which breast cancers are considered as HER2 positive. An important aspect of this is to understand if patients with breast cancer expressing HER2 but not amplified have the same outcome/treatment as patients with HER2-amplified breast cancer.

Neoadjuvant treatment: What is the rationale for using a combination of HER2 inhibitors?

Adjuvant treatment: What factors could explain the difference between the PHARE trial and the PERSEPHONE trial?

Metastatic disease: The authors should consider discussing first-line treatments versus second-line treatments. HER2 positive breast cancers have a high propensity to go to the brain (as mentioned in the Introduction) but there is no discussion on treatments of CNS metastases.

Mechanism of action of HER2 directed therapies and resistance: In this section, only the resistance to trastuzumab is presented. What about resistance to small molecule inhibitors?

Future directions: this section is too succinct and does not bring new insight into how treatment decisions could change.

Reviewer 2 Report

Review article by Patel et al., entitled ‘The changing Paradigm for the Treatment of Her2 Positive Breast Cancer’.

This article present a comprehensive review of the key studies, mostly clinical trials, investigating the different modalities of treatment for Her2 positive breast cancer since the advent of Her2 targeted therapies from Herceptin (trastuzumab) to newly introduced targeted therapies such as T-DM1 (trastuzumab emtasine). This review complements a very recently published review investigating the new avenues of anti-her2 breast cancer therapeutics option particularly in view of de novo or acquired resistance to actual her-2 therapies (Pernas and Tolaney, 2019 published in Ther Adv Med Oncol).

Comments

For the Persephone trial a more recent study from the same group mentioned has published an updated paper in The Lancet showing in addition to the non-inferiority of 6 months treatment compared to 12-month treatment with trastuzumab, less cardiotoxicity and fewer severe adverse events on 6-month trastuzumab. Earl et al., 2019 6 versus 12 months of adjuvant trastuzumab for HER2-positive early breast cancer (PERSEPHONE): 4-year disease-free survival results of a randomised phase 3 non-inferiority trial.

a few spelling mistakes were noted line 27 predilection instead of predeliction, line 218 resistance instead of resistnace.

Reviewer 3 Report

In this manuscript authors have reviewed the work on various treatments for Her2 positive breast cancer in clinic. 

Overall, this manuscript is well written and represents the ideas clearly and concisely. However, there are few areas which could be improved in this manuscript. 

Abstract: the abstract seems a bit weak in overall construction and content. Authors could indicate the advancement in this area towards the end of the abstract to highlight the success of current and upcoming therapies. 

Figure: the figure doesn’t do justice to the well written manuscript. Authors could bring more clarity to the figure to present better visuals. Additionally, there are various symbols which appear in the figure and need to be removed. What is ‘L’ in the figure (most can guess), but it should defined.  Perhaps adding a section or table on shortcomings of these therapies could enhance the manuscript overall.  Authors could add a figure or table to present various subtypes of breast cancer, highlighting Her2 subtype. Could also include the overall survival of these subtypes. Compare %survival of Her2 vs others. 

Define abbreviations at their first occurrence/ eg, pCR in line 59. 

Line 82- why two representations for “carboplatin” i.e. C in TCH and Cb in PCbH?